# Investigation of the Shot Size Effect on Residual Stresses through a 2D FEM Model of the Shot Peening Process

**Christos Gakias [1,*], Georgios Maliaris [2] and Georgios Savaidis [1]**

[1] Laboratory of Machine Elements & Machine Design, School of Mechanical Engineering, Aristotle University of Thessaloniki, 54124 Thessaloniki, Greece; gsavaidis@auth.gr

[2] Additive Manufacturing Laboratory, Department of Chemistry, School of Sciences, International Hellenic University, Aghios Loukas, 65404 Kavala, Greece; gmaliari@chem.ihu.gr

[*] Correspondence: clgakias@auth.gr; Tel.: +30-2310-996074

**Abstract:** Shot peening is a surface treatment process commonly used to enhance the fatigue properties of metallic engineering components. In industry, various types of shots are used, and a common strategy is to regenerate a portion (approximately up to 35% of the total shot mix weight) of used and worn shots with new ones of the same type. Shots of the same type do not have a constant diameter, as it is concluded by experience that the diameter variation is beneficial for fatigue life. The process of stochasticity raises the difficulty for the application of computational methods, such as finite elements analysis, for the calculation of pivotal parameters, for instance, the development of the residual stress field. In the present work, a recently developed plane strain 2D FEM model is used, which has the capability to consider various shot size distributions. With the aid of this model, it became feasible to study the effect of the shot-size distribution, its sensitivity, and to draw conclusions considering the industrial practice of using a mixture with new and worn shots. The diameter of these shot types differs significantly, and a used shot may have a diameter three times smaller than a new one. As concluded from the finite element results, which are verified from experimental measurements, a shot type with a larger diameter causes a wider valley in the stress profile, and the peak stress depth increases. Alongside the peak stress depth movement, with smaller shots, larger residual stresses are observed closer to the surface. Thus, the superimposition of many shots with variable diameters causes the development of a residual stress field with enhanced characteristics. Furthermore, this residual stress field may be further enhanced by adjusting or increasing the percentage weight of the used shots, up to ~50%.

**Keywords:** shot peening; finite element method; modeling; simulation; shot dynamics; residual stresses; shot distribution

## 1. Introduction

One of the most frequently used surface treatment procedures for improving the fatigue resistance of metallic engineering components is shot peening (SP). In this process, many small particles (shots) are fired at high velocity onto the surface of the processed part. The impacts plastically deform the surface of the treated component, yielding a compressive residual stress field at the surface that increases the resistance to fatigue crack initiation, decreases the crack propagation rate, and may even lead to the prevention of short cracks [1,2]. On the other hand, the SP process deteriorates the surface roughness in some cases [3,4], and, therefore, the fatigue life. The interactions between the above-mentioned effects are well described in [5–8].

The beneficial effect of the SP processes raised the interest of the industry for the development of processes compatible with the produced components. The large number of involved variables, such as shot velocity, shot size(s), shot size distribution, shot material properties, and impact angle, increase the complexity of the process design and the

necessary steps for the implementation of such methods. Because of this fact, industry's efforts rely on the experience acquired through experimental work. Although the use of simulation models based on the finite element method could provide precious insight, the large number of shots as well as their interaction with the surface of the processed component raise the computational effort up to very high levels, which seems to be the main limiting factor for the development of simulation models capable of describing the SP process with good accuracy. In the last decade, the continuous increase in computing power and the emergence of powerful and accurate commercial software helped to develop FE models that provide results with enhanced accuracy. Initial efforts focused on the simulation of single shots on bodies and the formation of the corresponding contact stresses as provided in, for example, [9,10]; the work of Zion and Johnston [11], which introduced a 2D axisymmetric FEM-based simulation model capable of considering various shot types; and Al-Hassani et al. [12], who considered multiple shots. Yang et al. [13] used a 2D plane strain FEA to analyze high coverage and Almen intensity. Extensive investigations followed to further study and model the physical phenomena of multiple shot impacts at surfaces using either 2D or 3D models [5,6,13–15]. In recent years, FEM-based simulation models capable of simulating the shot-peening process have been proposed that can explain and predict the correlation between the SP parameters and the process results. Significant progress regarding the modeling and simulation of SP processes focusing on residual stress development, the resulting surface roughness, and the cold work are described in [16–19].

Despite the progress made, crucial questions regarding the elastoplastic shot material behavior and the influence of shot size distributions have not been adequately addressed. Recently, a 2D FE model has been developed [20], using a plane strain approach similar to [13], capable of covering the aforementioned issues with sufficient accuracy. It considers elastoplastic material properties for the shots and real shot size distributions, a combination of properties that is also only present in [21], where the authors presented an extended study on the deviations of shot diameter. Moreover, due to its 2D dimensionality, it requires fewer computing resources and less execution time for model solution and evaluation of the appropriate process parameters. The further development of the specific model rendered feasible the conduction of simulations where different single and multiple shot size distributions are modeled. Therefore, the obtained results back up the practice applied in industry, where a certain percentage of worn shots are used with new ones. With the developed model, it was also possible to investigate the effect of the percentage of the worn shots on the developed residual stress field.

## 2. Finite Element Analysis Model

### 2.1. Short Description

The result of the meshing strategy is depicted in Figure 1.

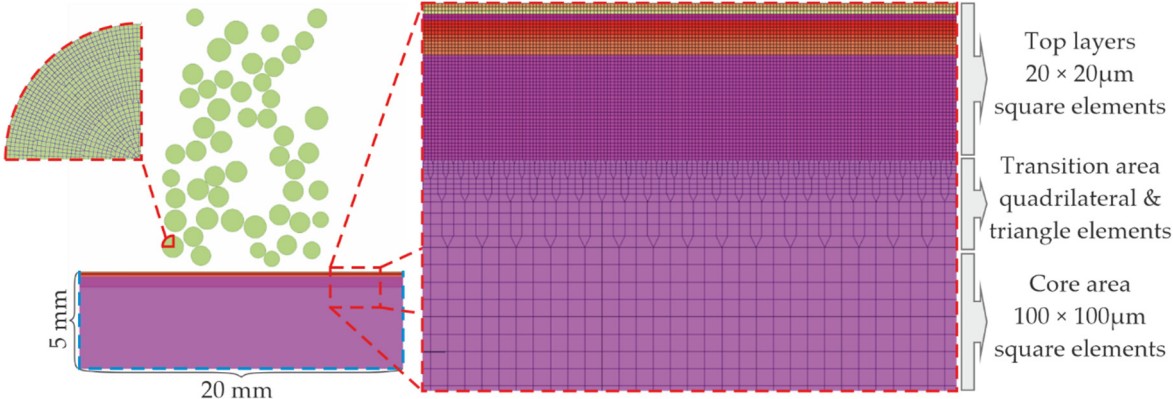

**Figure 1.** Specimen discretization strategy and essential dimensions [20].

While always keeping the required computational time under consideration, a relatively small part (compared to the real component geometry) is considered for the FEA model. In this study, the dimensions of the part under investigation amount to 20 mm in length by 5 mm in height (see bottom left part of Figure 1). These dimensions have been carefully selected to keep the solution time within acceptable limits and considering that the developed stress field must not reach the model's boundary. The cyan dashed lines on the bottom and left side of the model indicate that the nodes of the part's vertical and bottom edges are restrained in the horizontal (X) and vertical (Y) direction, respectively. The FEA model was discretized, applying a variable element size scheme; thus, the element size increased successively from the surface (very fine mesh) to the core area, as visible on the right-hand side of Figure 1.

The shots and the surface layers of the part have the same element length to improve the accuracy of the contact calculation algorithm. Pre-studies showed that an element length less than 20 μm may slightly improve the accuracy but increases the required computational time significantly. Further extensive information and thorough descriptions regarding the model characteristics are available in [20].

The elastoplastic material properties for both shots and specimen are depicted from [20] and presented in Table 1.

**Table 1.** Fundamental material values for specimen and shots.

| Parameter | Specimen: 51CrV4 | Shots |
| --- | --- | --- |
| Young's modulus (MPa) | 206,000 | 206,000 |
| Yield stress (MPa) | 1450 | 1500 |
| Ultimate tensile strength (MPa) | 1645 | 1700 |
| Strain hardening behavior | Multilinear elastic–plastic | Bilinear elastic–plastic |

Alongside the fundamental elastic–plastic material properties, a simple linear isotropic hardening rule was applied that also considers strain rate effects, as described in the study of Maliaris et al. [20].

Lastly, due to the generation of artificial stress waves, a special damping boundary condition was defined, which is applied on the nodes of the boundary of the specimen and permits the absorption of the artificial stress waves. Friction was neglected, since the assessment of the test run results revealed that the effect of friction on the calculated parameters is negligible, but the impact on solution time is countable.

### 2.2. Shot Size Information and Sieve Analysis

Each shot type used for industrial applications is characterized by a nominal diameter. The actual diameter value lies in a specific range that is determined by sieve analysis according to the SAE J444 standard [22]. Sieve analysis is a common practice to validate the size of each shot type, not only for metal blasting abrasives manufacturers, but also for the automotive and aerospace industry, as a control method for the quality of the peening process. Thereby, the shots pass through many sieve levels, the retained weight in each level is measured, and then the weight is converted to a percentage of the total. The percentage mass is usually expressed in a cumulative form according to SAE J444. The nominal sieve analysis data for the three different shot types S330, S390, and S460, which are widely used for automotive suspension components, are presented in Table 2. These data are available online from abrasives manufacturers like Kholee Blast [23] and Metal White [24].

**Table 2.** Sieve analysis data of brand-new shots according to SAE J444 (cumulative percentage mass for each sieve level).

| | Cumulative Weight Retained (%) | | |
|---|---|---|---|
| **Screen Size (mm)** | **S460** | **S390** | **S330** |
| 2.00 | All pass | | |
| 1.70 | <5 | All pass | |
| 1.40 | - | <5 | All pass |
| 1.18 | >85 | - | <5 |
| 1.00 | >96 | >85 | - |
| 0.85 | | >96 | >85 |
| 0.71 | | | >96 |

### 2.3. Industrial Shot Mixes

Feedback from industry and shot-peening experts has shown that operational shot mixes do not consist of brand-new shots. A shot-peening machine in industrial production is regularly replenished with new shots. Thus, an industrial shot mixture contains new, operational, and worn shots in a diameter distribution that is monitored over time. New shots are added when periodic sieve analysis shows a decrease in larger diameter shots, while the smaller shots (below a certain diameter) are sieved out of the mixture. To illustrate the differences in dimensions and shape, Figure 2 shows macroscopic photographs of the new, operational, and worn (but still in operation) shots from an industrial mix.

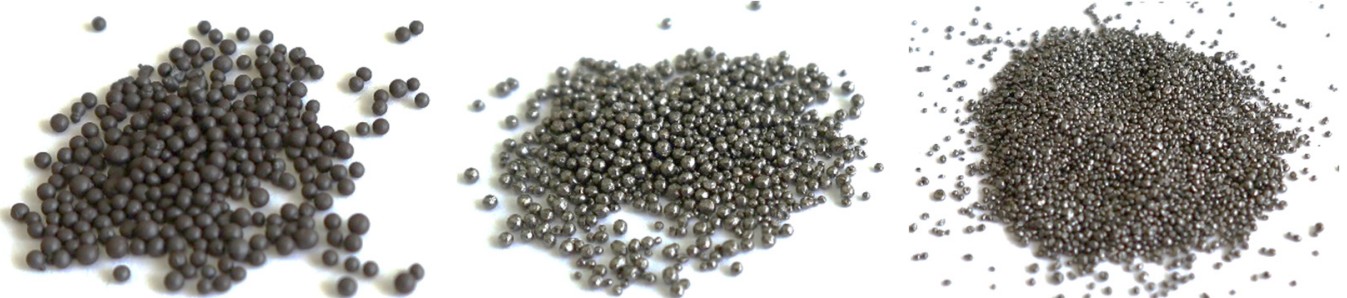

**Figure 2.** New (**left**), operational (**center**), and worn (**right**) shots.

A visual inspection of the operational shots revealed that despite the reduced diameter due to deformation and wear, their shape does not qualitatively differ from the shape of the brand-new shots. Therefore, all of the shots were considered spherical for the purposes of this study.

Table 3 gives the corresponding sieve analysis data from the industrial mix considered in the present study.

Figure 3 demonstrates an industrial shot mix distribution of operational shots (blue curve), compared to new/out-of-the-box S460 shots (black curve), as an example, expressed in both nominal and cumulative terms. This specific mix comes directly from a leaf spring manufacturer, who chooses to use S460 shots to refill the shot peeners in serial leaf spring production lines. It is obvious that the operational shot distribution differs significantly from the respective new shot counterparts, which shows that there is a substantial deformation of the shots during shot peening.

**Table 3.** Sieve analysis data of brand-new shots according to SAE J444 (cumulative percentage mass for each sieve level).

| | Cumulative Weight Retained (%) |
| --- | --- |
| **Screen Size (mm)** | **Industrial Mix** |
| 2.00 | 0.00% |
| 1.60 | 0.06% |
| 1.40 | 1.56% |
| 1.25 | 30.54% |
| 1.12 | 57.07% |
| 1.00 | 72.59% |
| 0.90 | 76.10% |
| 0.80 | 79.41% |
| 0.71 | 82.73% |
| 0.60 | 88.24% |
| 0.50 | 94.31% |
| 0.40 | 98.75% |
| 0.30 | 99.81% |
| 0.20 | 99.94% |
| 0.10 | 100.00% |

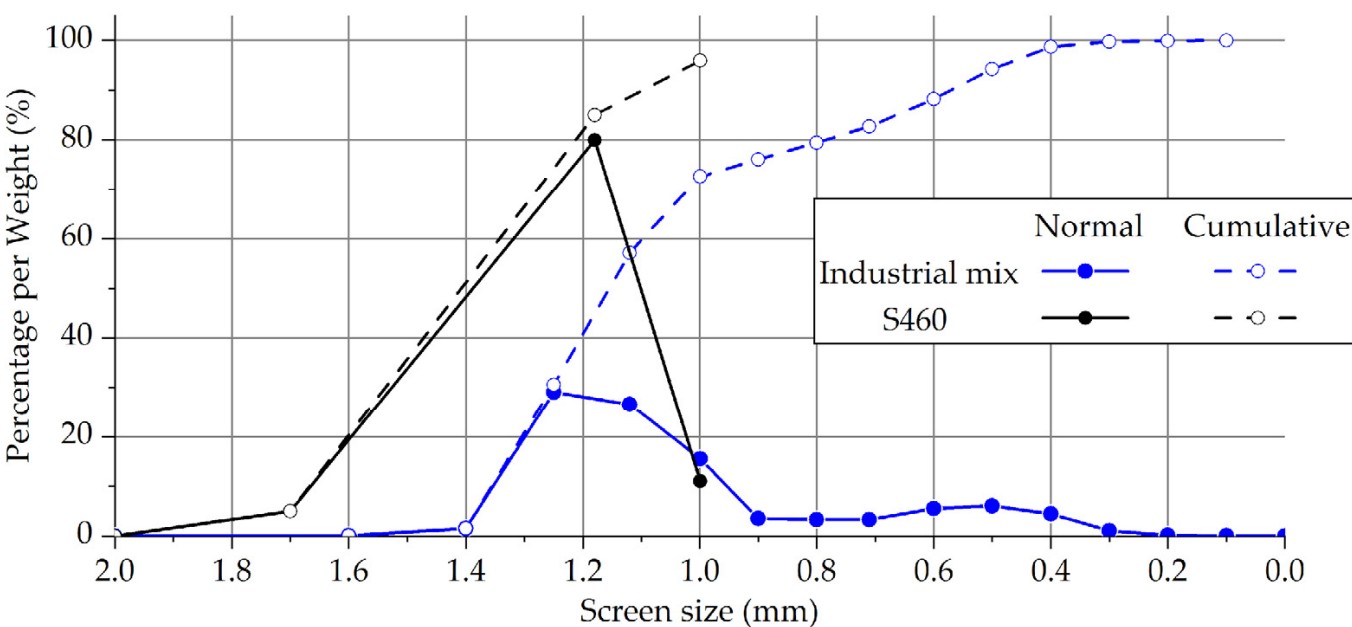

**Figure 3.** Normal and cumulative operational versus new S460 shot distributions per relative weight.

These curves are a more compact and practical way to express the data obtained from sieve analysis compared to the tabular form of Table 1. The Y axis represents the percentage of the weight of the shots that is retained on each sieve level. The screen size for each of those levels is located on the X axis. It is worth mentioning that an industrial sieve analysis procedure may not strictly follow the SAE J444 standard for the screen sizes, but it can be adapted to the needs of a specific industry.

It is easily observed that the curve of the industrial mixture has two local peaks. The first and higher one is located at a screen size of 1.25 mm, and the second at a size of 0.5 mm. Taking advantage of this fact, the operational shot distribution can be approached as two separate normal (Gaussian) distributions, using these peaks as a mean value. As a next step, using a basic statistical analysis, the standard deviation of each distribution can be extracted. These two main parameters of the Gaussian distribution (mean value and standard deviation) are crucial during the shot-generation procedure.

The same distributions are expressed in terms of relative (normal and cumulative) shot numbers in Figure 4. This transformation is essential to correlate the experimental and the computational results, since both the per relative weight and per relative number expressions must be closely followed. This is vital, as the numerical model generates a relatively limited number of shots (a few hundred) blasted over a small area of a few mm$^2$, in contrast to the laboratory or industrial conditions that are performed on a surface of several cm$^2$ being hit by many thousands of shots. Hence, this particular relation should be kept as close as possible to the experimental one, in order to achieve the same energy (i.e., momentum of shots = mass x velocity) being transferred at the same area (i.e., coverage).

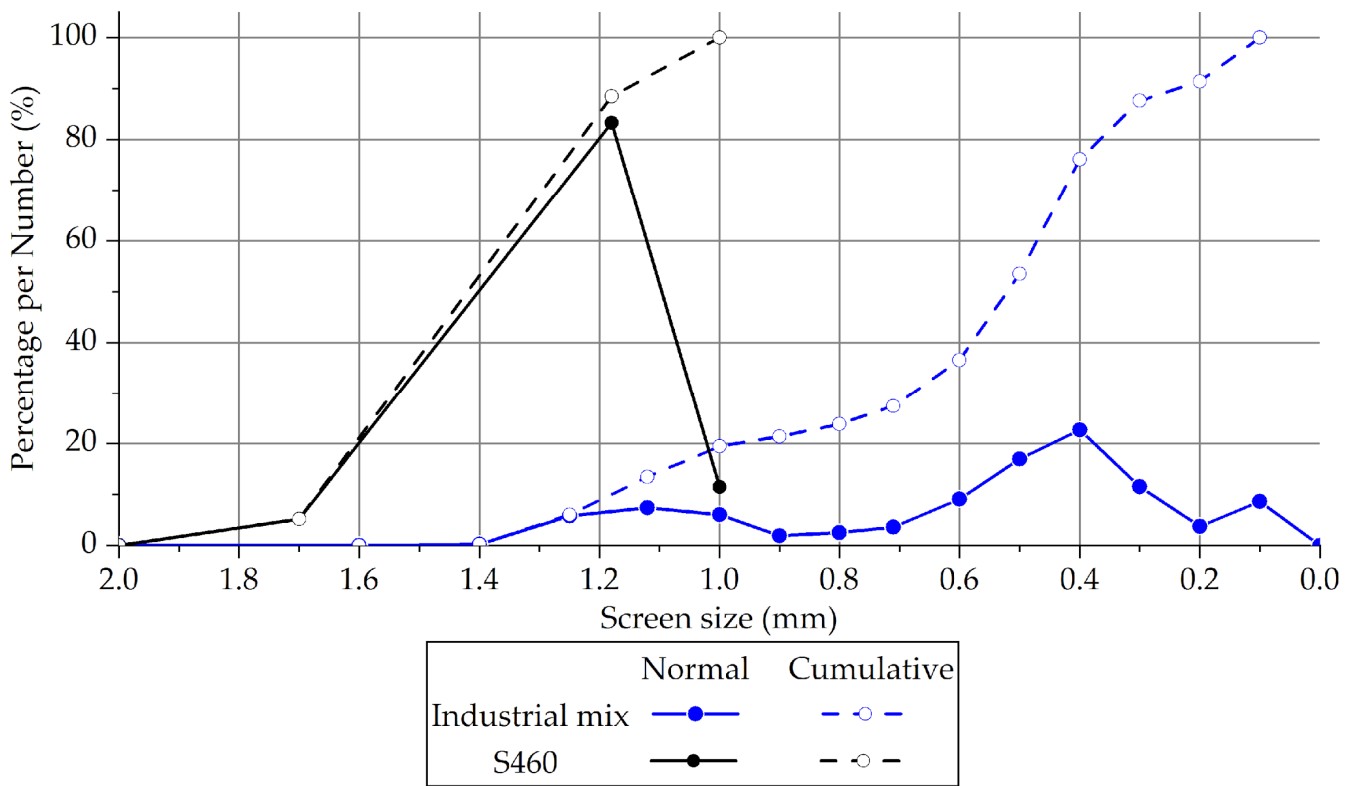

**Figure 4.** Normal and cumulative operation versus new S460 shot distributions per relative number.

The transformation of the relative mass (direct result of the sieve analysis) to the relative shot number considers the average diameter between two consecutive sieves as being the representative one for the respective class. For example, the relative mass of 29% sieved out of the 1.25 mm screen size in the industrial shot mix distribution is transformed into number of shots using the mean diameter of this grid and the previous one of 1.40 mm, resulting in a mean diameter of 1.325 mm. Furthermore, during the abovementioned statistical analysis, it is also vital to calculate the percentage occupied by each shot category (brand new or operational) on the total of the mixture, per relative number and per relative mass. For the current industrial shot mix, this percentage was calculated to ~35% of the total mass for the small (worn) shots.

### 2.4. Shots Generation Algorithm Development

To accurately model the brand-new and the industrial shot mix, a random sphere generation algorithm was developed that follows the sieve analysis data for these shot mixes. This algorithm was developed using the Python programming language alongside basic statistical packages. The implementation of this tool in the Application Programming Interface (API) of the ANSA© pre-processor led to the fast generation of the geometry of the spheres, an FE mesh creation, and the application of any boundary and initial conditions. The two main points that govern this algorithm are:

- The sphere generation, with a diameter that follows the given distribution(s).
- The random allocation of these spheres within a specified rectangular space domain.

Furthermore, this algorithm can check any intersection between spheres, consider any shot impact angle, and, most importantly, deviations and stochasticity can be considered for the given impact velocity and angle.

As for the first operating principle of the algorithm, the simple equation that is used for the diameter of a shot is the following:

$$\text{diameter} = \text{Gauss}(\mu, \sigma), \tag{1}$$

where Gauss indicates the Gaussian (normal) distribution and $\mu, \sigma$ are the main parameters: mean value and standard deviation, respectively. This technique is based on simple, fundamental statistics. Nevertheless, the developed algorithm includes the generation of shots, using more than one distribution, mixed in the same space domain. This is a simple and efficient way to accurately model a shot mix with a wide range and variety of shot diameters, as presented in Figures 3 and 4 above.

Regarding the second operating principle, the random position (X, Y, and Z center coordinates) of the shots in space in the case of the normal, vertical impingement can be described with the following equations:

$$X = \text{width} \cdot \text{uniform}(-1/2, 1/2) \tag{2}$$

$$Y = \text{height} \cdot \text{uniform}(0, 1) \tag{3}$$

$$Z = \text{length} \cdot \text{uniform}(-1/2, 1/2), \tag{4}$$

where uniform(a, b) is a uniform pseudo-random number generator in the interval [a, b] and is used for the random allocation of spheres in space. The variables width, height, and length stand for the given space domain width, length, and height for each of the three axes, respectively. For the case of an oblique shot stream, the X coordinate of each sphere is obtained by:

$$X = \text{width} \cdot \text{uniform}(-1/2, 1/2) + y/\tan(\text{angle}), \tag{5}$$

where angle refers to the impact angle of the shots. The abovementioned set of equations are based on the work of Hong [25]. The main difference between the current study and Hong's is the first operating principle, where the diameter of the shots is normally distributed. Nevertheless, the relative shot number and mass of the created shots are also checked as a final step.

As for the multiple created spheres, taking into account the out-of-the-box mixes, a Gaussian distribution was assumed, with a mean value equivalent to the nominal diameter and a standard deviation of 0.112 mm. The Gaussian distribution parameters were calculated using the sieve analysis data presented in Table 1, so the created spheres are in good agreement and satisfy the size tolerances according to the SAE J444 standard.

Furthermore, as clearly shown in Figures 5 and 6 below, the created spheres, according to the data of industrial mixes, accurately conform to both per weight and per number industrial shot distributions. These figures illustrate the comparison between the actual sieve analysis results of the operational mix and the corresponding classification data obtained from the developed algorithm.

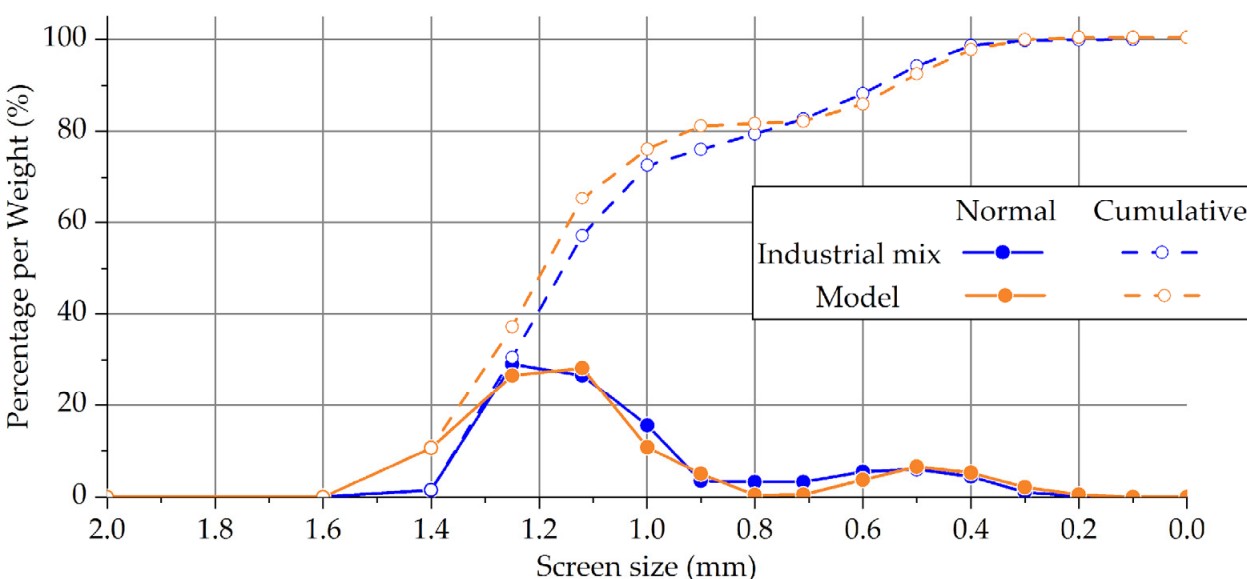

**Figure 5.** Comparison between sieve analysis and generated shots for the 2D FEA model; shot percentage per weight.

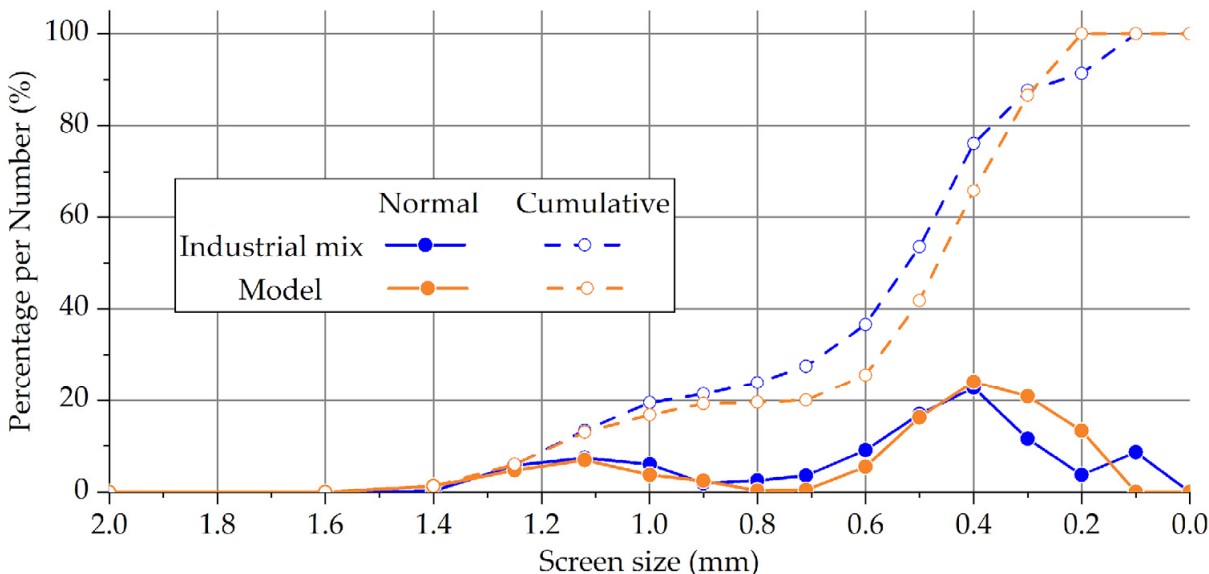

**Figure 6.** Normal and cumulative operational versus new S460 shot distributions per relative number.

As previously mentioned, the shot diameter distributions were approximated with two independent Gaussian distributions with mean diameters of 1.25 mm for the first peak corresponding to the new, recently added shots, and 0.5 mm for the operational shots. The statistical analysis was performed on the curve of the relative weight distribution, obtained directly from the sieve analysis. In addition, a standard deviation of 0.135 was calculated for both distributions.

In Figure 7a below, a close-up of the modeled industrial mix shots is presented and the large size deviations between shots are obvious. It can also be seen that there is a relatively larger number of small shots, which is also described in Figure 6 above. The industrial mix is compared with S460 mix, Figure 7b, and the shot diameter deviations are clearly visible.

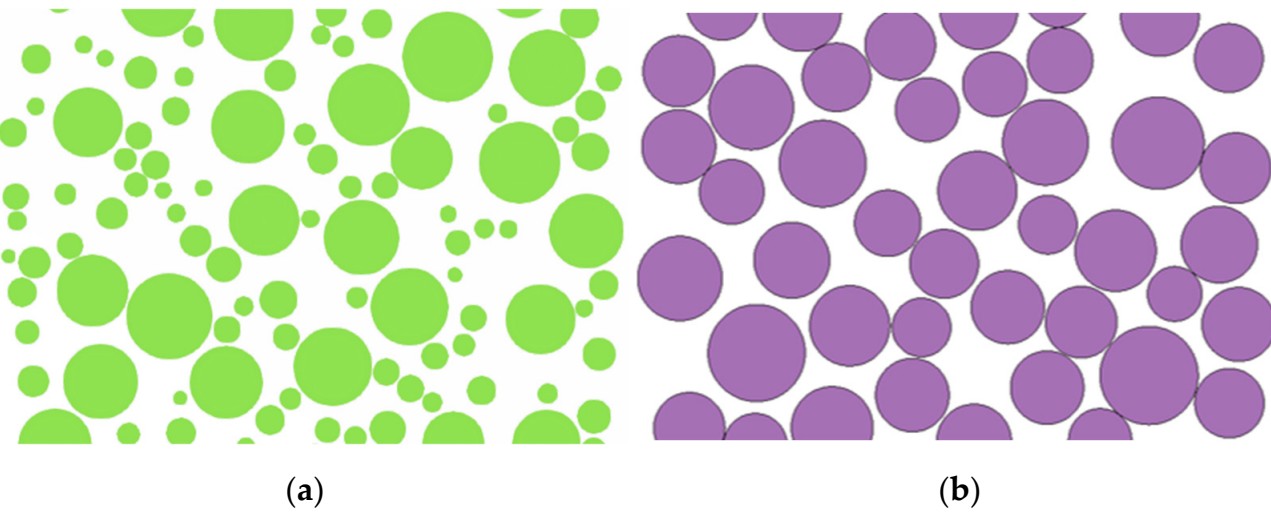

**(a)** **(b)**

**Figure 7.** Industrial shot mix (**a**) and S460 brand-new shots (**b**).

### 3. Results

Since the current study focuses on the investigation of shot size in the shot-peening process and its influence on the final surface properties, there will be an extensive presentation of the performed parametrical simulations. The first step for this study is a simple examination of the shot size using the three different shot types: S330, S390, and S460. The residual stresses for each case are compared, considering the size deviations. The second step is to study the influence of the stochasticity of the shot size. During this step, each shot type is studied separately to evaluate the effect of the shot diameter deviations. The final and most vital part is the further examination of the industrial shot mix, where parametrical simulations were carried out by adjusting the percentage quantity of the operational shots to investigate the results of the abovementioned variety in shot diameters and comparing them with a brand-new shot mix.

As for the results inspection, besides a simple, rough visual inspection of the stress field during the post-processing of the results, the compressive residual stress values on the surface for the whole peened area are averaged and plotted as a function of depth from the surface. Alongside those values, the standard deviation is also calculated, as a sign of stress field uniformity.

#### 3.1. Examination of Shot Size Effect

As a first step, a case study was conducted to examine the effect of the shot size, considering a distributed shot diameter for brand new shots, as presented in Table 1. Three simulations were carried out for S330, S390, and S460 shots, respectively.

Regarding the peening conditions, it should be mentioned that the total mass of shots that impacted the surface was equal to 1.5 g for the total peening length of 10 mm in each case with different shot types. For case equivalence, it was important that the same amount of energy was transferred to each specimen. This specific shot quantity was selected so that the final stress profile after the peening procedure was fully stabilized. The coverage of the surface was sufficient, and the stabilized stress profile was evaluated after many pre-studies, with multistep simulations, using a small number of shots on each step, until saturation occurred. The shots impacted the specimen vertically with a velocity of 77 m/s, which is a typical value for shot peening in leaf spring applications.

A visual example of the stabilized stress field of the specimen's surface peened with S460 shots is presented in Figure 8.

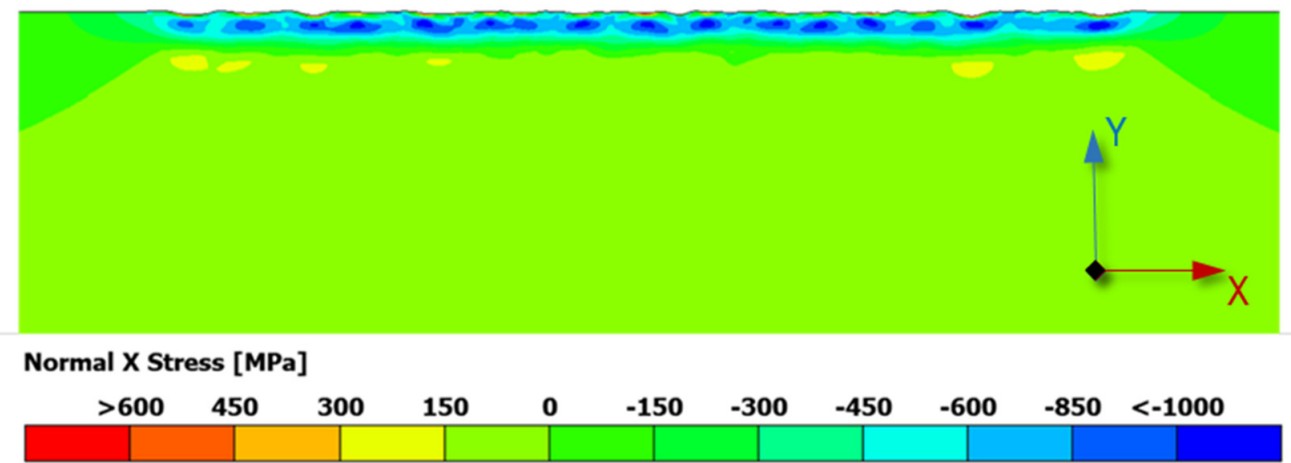

**Figure 8.** Stabilized compressive stress field after peening with S460 shots.

Figure 9 shows the calculated averaged residual stress profiles for brand-new shots of the three different shot types.

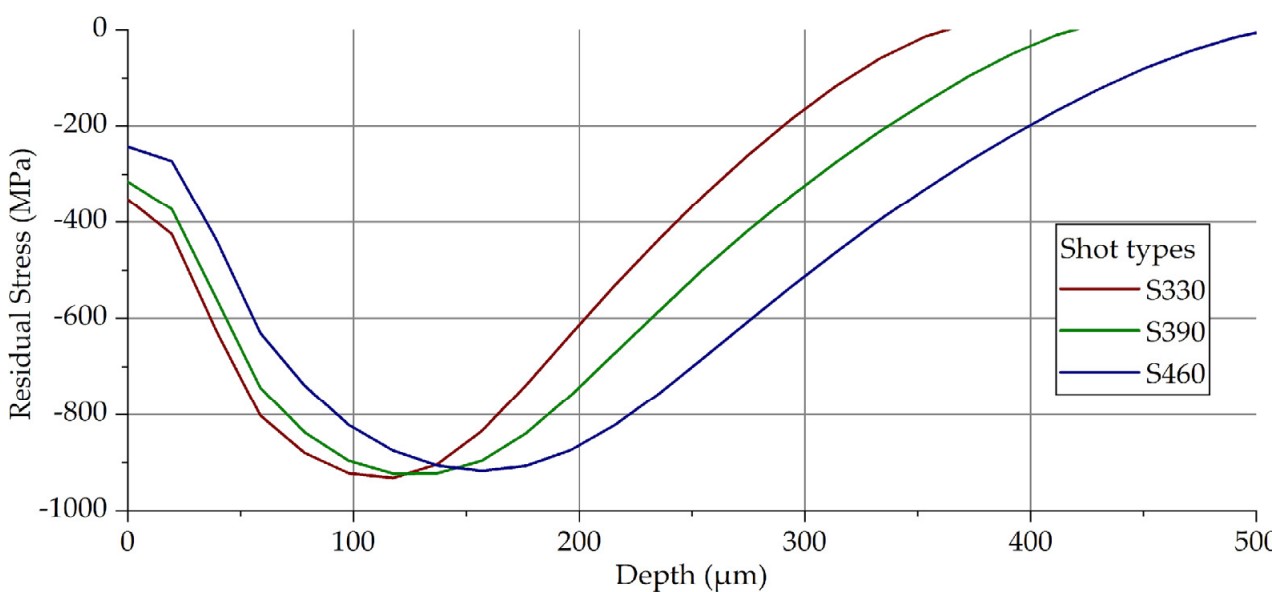

**Figure 9.** Averaged residual stress profiles for three different shot types.

The correlation between the shot type and the compressive residual stress profile, in terms of peak depth and surface stress value, can be summarized as follows: the larger the shot, the greater the peak depth and the wider the stress profile. Moreover, the peak stress value seems barely affected by the shot diameter. This conclusion is confirmed by the experimental studies of Ogawa and Asano [24] and it is easy to observe by comparing the purple curve (S460 type) with the red one (S330 type) in Figure 9.

Note that this peak value appears to vary slightly (a difference of 40 MPa between the red and the green curve), and this can be explained from the stochastic nature of the simulations (e.g., the randomly allocated spheres) and the averaging of the stress during post-processing. Hence, this difference can be considered negligible.

### 3.2. Influence of Shot Size Stochasticity

Proceeding to the next phase of this study, the actual influence of the diameter variation in a common shot-peening simulation was examined. As mentioned before, most studies suggested models with pre-allocated shots with a standard, uniform diameter that impact a surface. Each shot type was modeled separately. Random sphere batches for every type

were created, with or without a variable diameter. The distributed diameter sphere batches followed the sieve analysis data presented in Section 2.2, while the uniform diameter spheres were created with the nominal diameter of each shot type. Peening conditions were the same as the studies in Section 3.1 (vertical impact with a velocity of 77 m/s and equivalent mass for each case).

Figure 10 shows the S460 shots with variable diameter compared to the corresponding S460 shots batch with a uniform (constant) diameter.

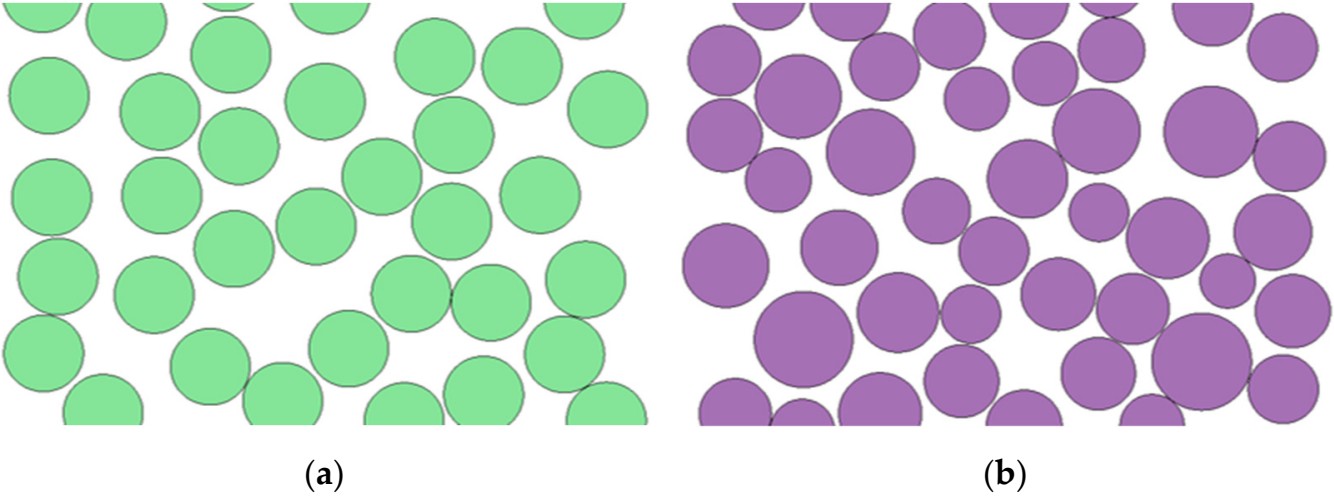

**(a)** **(b)**

**Figure 10.** Shots with constant diameter (**a**), and variable diameter (**b**).

The diameter variations are presented in Table 2. These deviations were reproduced using the developed algorithm, and the diameter of the modeled spheres lie between the following approximate ranges:

- **S330:** from ~0.7 mm to ~1.2 mm
- **S390:** from ~0.8 mm to ~1.42 mm
- **S460:** from ~0.9 mm to ~1.72 mm

The influence of size stochasticity on the residual stress profile regarding the shot types S330, S390, and S460 (all in brand-new condition) is revealed in the three diagrams of Figure 11.

For the three shot types investigated, it can be clearly recognized that diameter deviations affect the stabilized stress profile. The curves obtained with the distributed diameter shots models (solid lines) differ from those obtained with the constant diameter shot model (dashed lines). The differences at low depth values can be considered negligible because the residual stress values are almost identical. However, models with shots with distributed diameters tend to have a deeper final stress profile for every shot type, and slightly larger stress on the surface in most of them. The peak value does not seem to be affected by this (as also noted in Section 3.1).

These differences can be explained by the existence of smaller and larger spheres in each type. The number of these divergent-sized shots may be small, but sufficient to affect the stress profile, following the logic presented in Section 3.1. Larger shots induce a wider and deeper profile, while the smaller ones affect the area closer to the surface, leaving the peak stress value unaffected.

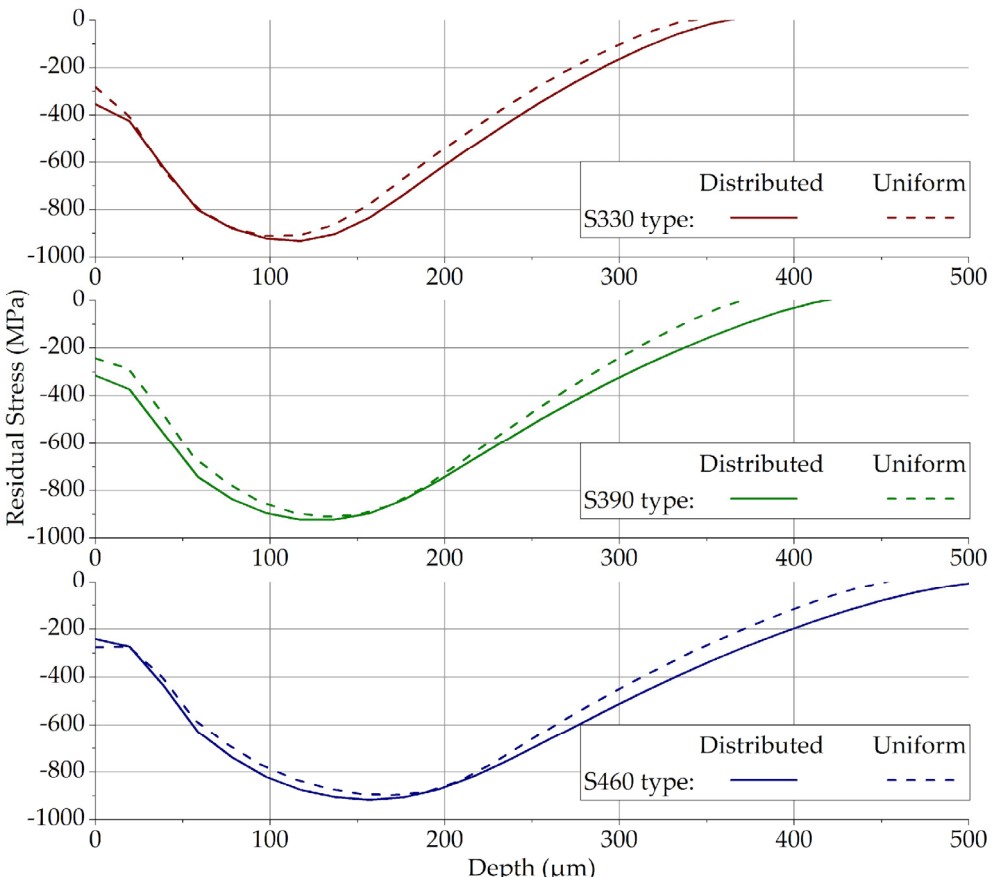

**Figure 11.** Comparison of residual stress profiles for three different shot types considering constant and variable shot diameter.

### 3.3. Industrial Shot Mix Investigation and Sensitivity

Proceeding to the third and most important part of this study, an extensive investigation was carried out regarding the differences between an industrial, realistic shot mix and a mix consisting of only out-of-the-box shots, as a more unrealistic scenario.

As mentioned above, an industrial peening machine is periodically fed with new shots, and rejects worn shots with diameters below a specific level. This feed-and-rejection process is mostly performed automatically. This specific mix was used for a series of experimental shot-peening studies, and the results of one of the many examined scenarios was reproduced using the developed model for the purposes of this study.

The experimental studies included the examination of different peening conditions on portions of leaf springs. Specifically, different materials, shot mixes, and velocities were examined. The scenario that was reproduced using the developed 2D model included the common 51CrV4 spring steel. Its mechanical elastic–plastic behavior was determined through tensile tests and was presented in [20]. It also included the shot mix that was presented in Section 2.3, and the peening velocity of 77 m/s under an impact angle of 10 degrees.

The residual stress profiles were measured using a Pulstec µ360s diffractometer, which uses the cosα method based on a full Debye–Scherrer ring [26–28]. The measurements are conducted lengthwise on the spring since normal shot peening produces an isotropic residual stress distribution in the surface plane [29]. The measurement interval was selected to be 50 µm in depth, up to approximately 480 µm, where the residual stresses nearly disappeared and the effect of the in-depth redistribution of stresses was considered. The measurement area was electropolished, so the induced stresses from the peening process remained unaffected.

A comparison of the calculated residual stress results and the measured stress profile for the industrial mix is presented in Figure 12. Therein, the orange solid curve represents the calculated averaged profile, while the shadowed area within the dashed lines represents the scatter of the calculated results.

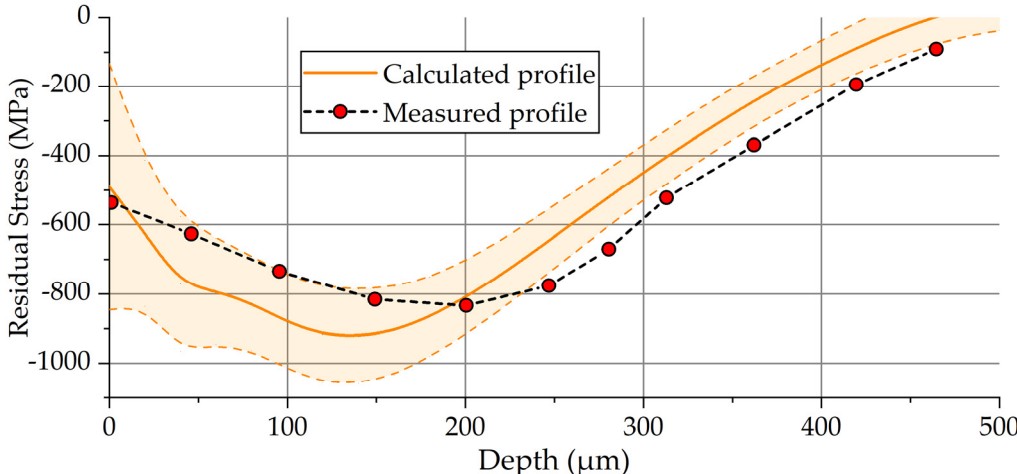

**Figure 12.** Comparison between industrial shot mix and experimentally measured residual stress profiles.

Considering that the 2D modeling represents a simplified simulation of shot peening, a rough comparison to the corresponding experimental results obtained by XRD reveals that the measured residual stress profile is in good agreement with the calculated one. In particular, in a depth up to 200 μm, the measured values lie near the upper bounds of the scatter band (standard deviation) of the simulated profile. An error of ~13% is visible in the peak stress value for the current peening velocity, and the shape of the profile is narrower.

Figure 13 presents the results from the sensitivity studies performed for the industrial shot mix in terms of the percentage mass of the recently added new shots and the worn ones. The percentage of worn shots in the current industrial shot mix is about 35%. The developed algorithm was used to reproduce two shot-type mixes, one with a corresponding percentage of around 15% and another of 50%. Again, for equivalency reasons, the same mass of shots was used in each study. The results seem to follow a specific pattern. The smaller (worn) the shots in the mix, the higher and wider the residual stress profile close to the surface. The surfaces' stress value ranges from ~400 to ~620 MPa, while the peak depth seems unaffected. Moving deeper from the surface, the less sensitivity the shot mix has, and the three profiles match.

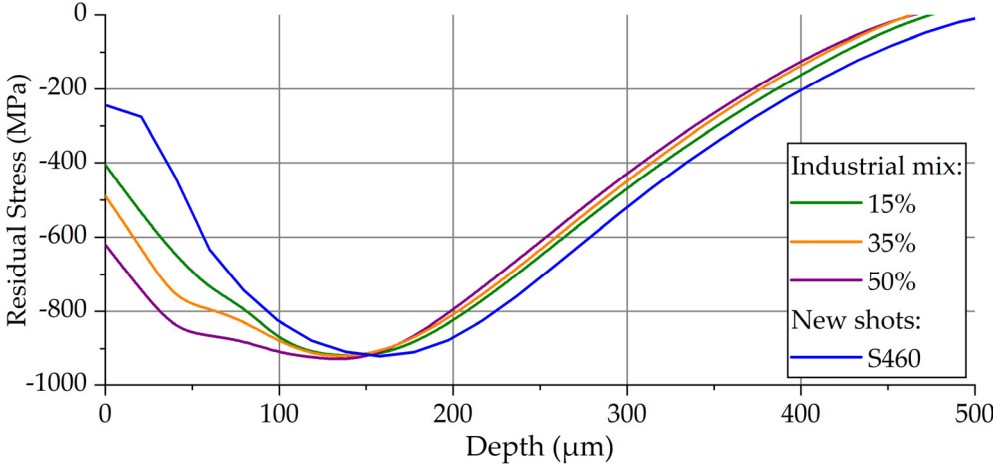

**Figure 13.** Residual stress profiles for various industrial shot mixes, alongside S460 shot mix.

## 4. Discussion

The studies presented in the previous section show how the result of the shot-peening process is strongly dependent on the shot mix. Changes in the shot mix may significantly alter the effect of the process on the peened component in terms of the developed compressive residual stresses. This observation is of significant importance, since studies—computational or experimental—of the shot-peening process should consider the changes and variations of the shot mix.

Proceeding to the leaf spring industry that provided the cases studied in this work, conventional shot peening and shot peening under high tensile stress (known as stress peening) are crucial milestones for the production and, consequently, the final quality of the products in terms of fatigue life. Since, in operational conditions, a leaf spring is loaded under a high-bending moment, the most intense tensile operational stress is in the upper surface. Thus, a larger residual compressive stress and a wider stress profile counteract the tensile stress obtained from that bending moment. This fact leads to the conclusion that smaller shots are intentionally kept in the production line peeners, and that they are not only beneficial for the production costs, but also the quality of the final product. Furthermore, an increase in the number of smaller shots may further optimize the fatigue life of a leaf spring. Figure 13 illustrated an improved residual stress profile, which may be more beneficial for fatigue life as the residual stress values were higher closer to the surface. The 50% increase of the amount of reduced-diameter shots showed an improvement of the residual stress profile, revealing space for further optimizing the fatigue life of the product. Scuracchio et al. [30] experimentally demonstrated the beneficial effect of secondary peening with smaller shots on the fatigue life of leaf springs, verifying the outcome of the calculation results of the present study.

However, an increase in the percentage of the mass of operational shots in an industrial mix may not be beneficial for the residual stress profile. After a specific percentage threshold, brand-new shots may not be sufficient, and the stress profile will become narrower and converge to a lower peak depth. The investigation of this percentage threshold is outside the scope of these studies.

## 5. Conclusions

Based on the results of the present studies, it can be safely concluded that deviations in the diameters of the shots can significantly affect the final compressive residual stress profile. In addition, when a shot mix consists of spheres with large deviations in diameters and different shot types, such as a typical industrial mixture, these deviations are important to consider. Specifically, it has been proven that the existence of used shots in a shot mix, in a percentage up to 35% per weight, reduces the residual stress peak depth up to ~12.5% and increases the surface value up to ~50%. Furthermore, a further adjustment of $\pm 15\%$ of the percentage weight of the used shots increases or decreases the surface stress value up to ~20%, respectively. The assumption of uniform shot diameter equal to the nominal value, which often occurs in simulation processes, may lead to large errors. This conclusion is considered important and useful for future research on the optimization of such processes.

Using this 2D simulation approach, an accurate qualitative assessment of the residual stress profiles after shot peening can be promptly achieved, as well as with useful information for the behavior and sensitivity of the process, in minor or major condition changes. Therefore, the shot-mix adjustment presented in Figure 13 and discussed in paragraph 4 might be considered as a suggestion for an improvement of the process in industrial conditions.

**Author Contributions:** Conceptualization, C.G. and G.S.; methodology, C.G. and G.M.; validation, C.G., G.M. and G.S.; investigation, C.G.; resources, G.S.; writing—original draft preparation, C.G., G.M. and G.S.; writing—review and editing, G.S.; supervision, C.G. and G.M.; project administration, G.S.; funding acquisition, G.S. All authors have read and agreed to the published version of the manuscript.

**Funding:** This research was funded by Research Fund Coal & Steel, grant agreement number 799787. The authors would like to gratefully acknowledge the Research Fund Coal & Steel.

**Acknowledgments:** BETA CAE is gratefully acknowledged for the provision of ANSA and META software.

**Conflicts of Interest:** The authors declare no conflict of interest. The funders had no role in the design of the study; in the collection, analyses, or interpretation of data; in the writing of the manuscript, or in the decision to publish the results.

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
