# Peer review of "Investigation of the Shot Size Effect on Residual Stresses through a 2D FEM Model of the Shot Peening Process"

_metals, doi:10.3390/met12060956_

Round 1

Reviewer 1 Report

The study of 2D FEM model of the shot peening process is not a novelty topic.

The Discussion in the paper is short of theoretical analysis. And the Conclusions is also improper.

Reviewer 2 Report

The authors studied the effect of shot size distribution on the residual stress of the shot peening process and found a difference in using uniform and distributed shot size. They also studied the residual stress behavior with varying shot sizes. The results are of interest to both the simulation and experiment community, and the manuscript is well written. I thus suggest accepting this manuscript, after addressing the following issues:

  1. “The impacted surface layers exhibit pronounced strain hardening, increased local yield stress and, thus, resistance to fatigue loading.”

--- This statement is not accurate and could lead to a misunderstanding. The impacted layer is just a gradient nanocrystalline layer. The top nanocrystalline layer results in high yield strength, and the coarse-grain core promotes strain hardening. It is the synergistic effect of the above factors that lead to the high yield strength, and strain hardening, and thus an increased resistance against mechanical damage. The statement saying that the impacted surface layer promotes high strength and strain hardening etc is too simplified and misleading.

“On the other hand, the SP process deteriorates the surface roughness and therewith the fatigue life”

“deteriorates the surface roughness” is not accurate.

  1. The results of the current work should be summarized there, e.g. a shot type with a larger diameter causes a wider valley in the stress profile, and the peak stress depth increases. Alongside the peak stress depth movement with smaller shots, greater residual stresses are observed closer to the surface...

Reviewer 3 Report

The manuscript cannot be accepted in its present form; it needs a major revision. The following critical points should be taken into consideration.

Abstract

The results in the Abstract should also be value results (i.e. 5% increase or 10% decrease by changing parameter.) It is not enough to write "As it is concluded from the finite element results, which are verified from experimental measurements". The important thing is why it is verified, or what are the reasons for the deviations?

Introduction

The authors could not indicate the gap in the literature well. It should also be improved. There are also 2D FEM-based simulation models capable to consider various shot types in the literature. What is the novelty of the study?

Finite Element Analysis Model

".....hereby the element size increased successively from the surface (very fine mesh) to the core area". Why is it only "very fine mesh"? It cannot be used in any scientific contribution? The authors should perform mesh sensitivity analysis to determine the optimum mesh size. 

In Table 1, Fundamental material values for specimens and shots are not adequate to perform the FE modeling of any explicit dynamic analysis. Where are the Johnson-Cook parameters of the specimen which will be plastically deformed?

How did the authors determine the experimental parameters? If from any literature studies, please cite them.

Results and Discussion

Numerical FE results are unfortunately not confidential due to the above-mentioned deficiencies in FE procedure even if experimental results are well presented.

Discussion

Thanks to the authors for a good discussion. However, general results sections should be improved

Conclusion

The conclusion section should be expanded with also numerical results (%X increase or decrease by this effect, etc).

General 

The paper should be proofread and checked based on typo errors.

Author Response

I would like to inform you that the text was proofread and improved.

Reviewer 4 Report

Investigation of the shot size effect on residual stresses through a 2D FEM model of the shot peening process (review)

The paper studies the influence of particle size in a shot peening process on the distribution and magnitude of residual stresses. The authors built a 2D numerical model of a specimen and developed a strategy for its loading by multiple particles (shots) that can have different sizes and impact the specimen from different angles. They also analysed realistic industrial shot mixtures and studied the different amounts of new shots added to the mixture. As a result, they showed the influence of different industrial mixtures on residual stress distribution. The work has no significant scientific value or contribution but is still an interesting analysis for engineering and industrial practice.

The central issue of the paper is the considered 2D FEM model for the simulated shot peening process, which is physically incorrect. The model can give some qualitative insight into the process for practical reasons, but this must be fairly discussed. Firstly, whether the 2D model is considered plane stress or plane strain must be explained. And secondly, there is no attempt to confirm that the model gives similar distribution as the correct one. I suggest the following: Upgrade the introduction and/or the modelling section with the literature review about shot peening modelling because, in the current version, references are somewhat old or not relevant to this issue. And the most important: they are not compared to the modelling technique used in the paper. Several papers can serve for comparison of modelling techniques, like Gariépy et al.: Simulation of the shot peening process with variable shot diameters and impacting velocities (2017) which also analysed shot size effect, or Starman et al.: Differences in phase transformation in laser peened and shot-peened 304 austenitic steel (2020) where numerical aspects of the modelling were discussed or Bagherifard et al.: On the shot peening surface coverage and its assessment by means of finite element simulation: A critical review and some original developments (2012) where they presented the approach to model a surface coverage. Of course, several other papers can be used to compare the modelling principle.

In the modelling section, I suggest comparing residual stresses vs depth of a single shot analysed with three models (for a single set of input data): plane stress, plane strain and axisymmetric model. Your existing model can be easily converted to the other two, so there is no considerable effort to compare the residual stress distributions of all three models. Based on their comparison, Figure 13 can also be perhaps additionally commented on.

This was a general remark; the additional specific ones are as follows:

  1. Table 1, "Modelling approach": Did you mean hardening behaviour?
  2. Figs 3 & 4: In Fig. 3, there is a larger peak for an industrial mix at 1.2 mm and a smaller one at 0.5 mm, whereas, in Fig. 4, the height of both peaks is replaced. Can disintegration of shots into smaller pieces explain this? Have you observed crushed particles? Can you also comment on the shape of these shots in the industrial mix, please? Namely, they are all modelled as spheres.
  3. Sect. 3.1: "1.5 gm": How to interpret this unit? Did you mean g/m?
  4. Sect. 3.1: "stabilized stress profile": If the stress profile is stabilized, i.e. it does not change with the additional shots, why did you emphasize the equivalency about the same amount of energy if nothing changes after some point?
  5. Sect. 3.2: Please, add the diameter variation (range).
  6. Sect. 3.3: Measured stresses are mentioned, but there is nothing about measurements, primary data, how they were performed, etc. I suggest adding a paragraph about measurements.
  7. Figure 14: The results indicate that a 50% industrial mix gives the most considerable compressive residual stresses, so a mix with fewer new shots is considered better - such an explanation is also found in the discussion and conclusion section. However, if we extrapolate, one might think that 100% of the industrial mix is the best, so no new shots are needed. Please, discuss.
  8. Proofreading of the text is strongly recommended.

I suggest improving the text before resubmitting the manuscript.

Author Response

(The authors gave the same response as above.)

Round 2

Reviewer 1 Report

English language is poor. The The innovation of article content is insufficient.

Reviewer 3 Report

The authors revised the manuscript according to my comments. The paper can be accepted as it is.

Author Response

We would like to thank you for your notes and comments.

Reviewer 4 Report

Investigation of the shot size effect on residual stresses through a 2D FEM model of the shot peening process (review)

The authors have addressed and resolved all the most important issues in the revised manuscript, thus I recommend it for publication.

Author Response

Thank you for your accurate and useful comments.